# Childhood Maltreatment and Adult Work Absenteeism: Work Meaningfulness as a Double-Edged Sword

**DOI:** 10.3390/ijerph21040451

**Published:** 2024-04-07

**Authors:** Tamar Icekson, Avital Kaye-Tzadok, Aya Zeiger

**Affiliations:** 1School of Behavioral Sciences, Peres Academic Center, Rehovot 7610202, Israel; 2Department of Management, School of Education, Ben-Gurion University of the Negev, Beer-Sheva 8410501, Israel; 3The Lior Tsfaty Center for Suicide and Mental Pain Studies, Social Work Department, Ruppin Academic Center, Emek Hefer 4025000, Israel; avitalk@ruppin.ac.il; 4Coller School of Management, Tel Aviv University, Tel Aviv 6139001, Israel; ayazeiger@mail.tau.ac.il

**Keywords:** work meaningfulness, work absenteeism, job demands–resources, childhood maltreatment, adult survivors, helping professionals

## Abstract

The adverse impacts of childhood maltreatment (CM) on an individual’s health and economic welfare are widely recognized, yet its occupational and organizational effects remain less explored. Employee absenteeism, known as absenteeism, is often a sign of workplace maladjustment and may be linked to a history of CM. Some individuals in the helping professions, who exhibit a strong sense of purpose in their employment and pursue it in demanding environments, are CM survivors. This study investigates whether a heightened sense of meaningfulness in their work is associated with increased absenteeism among this subgroup. We recruited 320 helping professionals from a variety of social and mental health settings, one third of whom reported experiencing CM. As hypothesized, CM was positively correlated with work absenteeism. Furthermore, the relationship between work meaningfulness and absenteeism was moderated by their CM history: among those with CM experiences, greater work meaningfulness was associated with higher absenteeism rates. Our findings highlight the possibility that work meaningfulness may operate as a double-edged sword, and the importance of better understanding the challenges that high-functioning survivors of CM face within organizational contexts.

## 1. Introduction

Childhood maltreatment (CM) is a commonly encountered traumatic event, with extensively documented physical, psychological, and economic adverse effects [1,2,3]. Less is known, however, about CM’s occupational and organizational long-term consequences. Several studies which have explored the challenges CM survivors face at work suggest that they may represent a vulnerable subgroup of employees [4,5]. Despite their contribution, many studies conducted among working CM survivors have focused on samples of blue-collar, low-income employees [6]. However, the working conditions and work environments of various types of professionals may have different characteristics. The aim of this study is to address this gap by focusing on white-collar, high-functioning professionals who have suffered from CM. Moreover, many studies have utilized qualitative methods focusing mainly on the subjective experiences of small samples of survivors in the workplace [7,8,9,10]. While this approach is valuable, a better understanding of workplace behaviors calls for the use of larger samples and quantitative examinations. Such an exploration will aid our understanding of the relationship between CM history and workplace outcomes, such as work absenteeism.

Work absenteeism is one of the strongest indicators of employee maladjustment and studies have documented its negative impact on work outcomes. Performance, innovation, employee turnover, and a firms’ productivity have been shown to be affected by work absenteeism [11]. Considering these costs, interest in absenteeism’s causes and related processes continues to grow [12,13]. According to the conservation of resources theory (CORs) [14], people seek to obtain, preserve, and protect resources, while a lack of resources can lead to stress and maladjusted work behaviors, such as absenteeism. The Job Demands–Resources (JD-Rs) model, which builds on the CORs theory and broadens it, suggests that absenteeism increases when individuals experience persistent and high demands, while lacking the adequate resources to address or buffer them. According to the JD-Rs model, demands as well as resources may be personal or work related [15].

Research on work absenteeism conducted within the framework of the JD-Rs model has focused on the role of job demands and resources, with most studies focusing on the influence of social, contextual, and organizational demands (for a review see [12]). Less attention has been given to personal demands, despite their significant role in shaping employee functioning at work, especially attendance behaviors [14]. According to the life-course perspective, facing negative life events, including those at early ages, may play a role in developing later workplace maladjustment [16].

One major personal risk factor for work maladjustment, rarely explored in the context of work absenteeism, is a history of childhood maltreatment. Due to their ongoing struggle caused by their histories, one might assume that CM survivors will miss more days at work compared to their non-maltreated counterparts. However, the empirical support for the relationship between CM and work absenteeism is scant [17,18,19]. Thus, the first aim of this study is to address this gap by studying the relationship in the workplace between CM as a personal demand and work absenteeism.

Work absenteeism may also be related to a sense of meaningfulness at work. Work meaningfulness has been the focus of psychological scientific inquiry for decades and has been considered to be positively and consistently associated with individual and organizational benefits, such as commitment, satisfaction, and performance (for reviews see [20,21]). Alongside the accumulating knowledge regarding the benefits of work meaningfulness, recent reviews have called for adopting a more critical approach to understanding its impact [22]. Some scholars have suggested that an enhanced sense of value, importance, and investment in their work may actually have a negative impact on employees who are psychologically vulnerable, especially at times or in situations that are highly demanding [23,24,25]. Following this line of thought, the current study suggests that CM survivors who work in the helping professions may represent such a group.

The possible vulnerability of these employees can be explained by several mechanisms. First, the percentage of CM survivors in the helping professions is higher than in other professions, and higher than in the general population [26,27]. Second, survivors of CM frequently characterize their occupations within the helping professions as “callings” [28,29]. While a sense of calling at work is intuitively beneficial, and indeed quite often it is [30], high levels of a sense of calling have also been found to be associated with negative occupational outcomes, especially for workers in very demanding roles and working environments [31,32,33,34,35].

Third, work in the helping professions is often characterized by emotional pressures, irregular shifts, heavy workloads, and work–home conflicts [36,37]. Fourth, working as a helping professional may expose employees to clients’ traumatic experiences, thus increasing the risk of burnout and secondary traumatization [4,37]. For CM survivors, these experiences may trigger painful memories [38], making them more vulnerable, especially when faced with traumatic events at work [5].

Overall, work meaningfulness typically functions as a personal job asset for non-maltreated employees. However, for CM survivors employed as helping professionals, work meaningfulness may be associated with increased job absenteeism. Thus, the second aim of this study is to explore the possible moderating role of CM history on the relationship between work meaningfulness and absenteeism among helping professionals.

### 1.1. Childhood Maltreatment and Workplace Absenteeism

CM has been defined by the World Health Organization [39] as “abuse and neglect that occurs to children under 18 years of age, which includes all types of physical and/or emotional ill treatment, sexual abuse, neglect…and which results in actual or potential harm to the child’s health, survival, development, or dignity in the context of a relationship of responsibility, trust, or power”. A comprehensive meta-analysis of the prevalence of CM in Europe and North America found that 23.5% of individuals reported experiencing at least one type of childhood maltreatment; 19% in Europe and 35% in North America reported experiencing two or more types of CM [40].

Experiencing CM has been found to be related to a wide array of negative outcomes in several areas of adult life. First, experiences of abuse and neglect in childhood have well-documented adverse consequences on physical health [1,2]. For example, cardiovascular disease, certain types of cancer, stroke, asthma, chronic obstructive pulmonary disease, kidney disease, diabetes and obesity have all been found to be related to CM [41]. Second, an ample number of studies have documented the adverse mental health consequences of CM, such as depression, anxiety, and post-traumatic stress disorder [6,42]. Scholarly work has offered insight into the mechanisms which explain these adverse psychological and physiological outcomes. For example, a recent literature review suggests that CM may impact neurological and physical functioning causing a higher vulnerability for affective disorders [43].

While there are studies that have contributed to a better understanding of the physical and psychological consequences of CM in adulthood, less is known regarding its occupational and organizational outcomes. CM has been found to be related to economic outcomes in adulthood, including occupational instability, underemployment, and having a low income [3,18,44,45]. However, the explanatory mechanisms regarding workplace maladjustment among survivors are not as well developed compared to the studies on its psychological and physiological outcomes [43]. The few studies which have been conducted on the workplace experiences of CM survivors suggest several possible mechanisms. For example, some studies point to increased relational challenges, such as a difficulty getting along with co-workers and supervisors [8,9,10]. Other studies report that CM survivors may be more vulnerable to victimization in the workplace, such as bullying or sexual harassment [7,46].

The exploration of work-related outcomes among CM survivors has documented a higher vulnerability to stress at work. For example, studies conducted among professionals with a history of CM have shown higher acute stress responses [5], lower levels of compassion satisfaction [4], and, in some cases, higher levels of emotional exhaustion, secondary traumatic stress, and job burnout [5,47,48,49].

Most germane to this research are the very few studies which have explored the relationship between CM and work absenteeism. Work absenteeism has been defined as an “individual’s lack of physical presence at a given location and time when there is a social expectation for him or her to be there” [50] (p. 263). Work absenteeism serves as a robust indicator of employee maladaptation, with recent research highlighting its detrimental effects on various work-related outcomes [11]. Given the substantial implications of absenteeism, there is a growing interest in elucidating its causes [12,13].

As posited by the Job Demands-Resources (JD-Rs) model, demands and resources can originate from personal factors or from the work environment [15]. Research investigating work absenteeism within the JD-Rs framework has predominantly focused on the impact of job-related demands and resources, with less attention being given to personal demands [12], despite their potential to contribute to explaining the phenomenon [51]. Considering the long-term deleterious consequences of CM, especially when coping with more physical and mental health challenges, a CM history may be positively related to work absenteeism. Indeed, in studies, among non-maltreated employees, symptoms of depression and anxiety predicted work absenteeism [52].

Despite these promising avenues, only a handful of studies have directly explored the relationship between CM histories and later work absenteeism or tried to uncover some of the explaining mechanisms. The scant literature on the topic suggests that work absenteeism may be higher among CM survivors [17,18,19]. In recent years, affective disorders [17], neuroticism [53], perceived work stressors [53], subjective social status [54], and trait anxiety [54] have been suggested as possible mediators between CM histories and later work absenteeism. The first goal of the current study was to explore the relationship between a history of childhood maltreatment and work absenteeism. Building on the aforementioned literature, it was hypothesized that absenteeism levels among maltreated employees will be higher compared to those among non-maltreated employees (H1).

### 1.2. Work Meaningfulness as a Double-Edged Sword: The Role of Childhood Maltreatment History

Above and beyond the direct associations of experiences of abuse and neglect in childhood with adult work absenteeism, the existing literature suggests that a CM history may attenuate the relationship between personal resources, such as meaningfulness, and work behaviors, such as absenteeism. CM survivors working as helping professionals represent a subgroup of vulnerable, highly invested employees for whom work meaningfulness may have a detrimental impact.

Work meaningfulness generally refers to the experience of work as holding a positive meaning and significance for individuals [55]. In its broadest sense, meaningfulness pertains to the experience of work being of value and worthwhile. Two sub-dimensions seem to be pivotal in most scholarly definitions: the work’s contribution to the greater good or prosocial objectives; and self-realization, denoting a sense of autonomy, authenticity, and self-expression within the work context [55,56,57,58].

Work meaningfulness is usually considered to be a prominent personal resource at work [59]. Personal resources were first conceptualized as aspects of the self that contribute to one’s adjustment [60]. Later definitions of personal resources referred to “assets that are valued by a person and that are directly available to improve effective functioning in specific domains” [61] (p. 2). Considering its nature, it is unsurprising that a plethora of studies have documented the beneficial work-related outcomes of meaningfulness, including enhanced commitment, engagement, satisfaction, and performance, as well as diminished leaving intentions [20,21]. One prominent work-related outcome which has been less studied in regard to work meaningfulness is work absenteeism. Following the JD-Rs model, personal resources have the potential to reduce work absenteeism [15,62]. Indeed, work meaningfulness, as a central personal resource, has been found in a few studies to be negatively related to absenteeism [56,63].

Despite the increasing evidence which suggests that work meaningfulness plays a beneficial role in predicting the desired work outcomes, questions arise regarding the conditions, contexts, and subgroups for which adverse ramifications of work meaningfulness may develop [20,21,23]. According to Bailey and colleagues [22], there is a paradox between the need of individuals to find meaning at work and the adverse consequences this effort may bring about: “individuals have an innate drive to seek out meaningful work to satisfy their inner needs, yet this same drive can push them to harmful excesses” (p. 489).

Several empirical attempts lend support for Bailey et al.’s claim [22] and try to shed light on its “dark side”. In one of the first explorations of meaningful work as a double-edged sword, Bunderson and Thompson [31] found that zookeepers who perceived their jobs to be highly meaningful (i.e., as a calling) displayed a greater inclination to view their work as a moral obligation and to sacrifice pay, personal time, and comfort for it. Similar results have been found concerning an enhanced willingness to overwork and erratic job behavior among international aid workers [32]. In a similar vein, it was recently found that the negative impact of perceived discrimination on job satisfaction is weaker among those who view their work as a spiritual calling (i.e., having a high sense of work meaningfulness), a mechanism that may allow discrimination in organizations to potentially go unaddressed [64].

The second stream of research points to the degree of work meaningfulness as a key factor of strain, suggesting that elevated levels of work meaningfulness, in certain contexts, may lead to an erosion of wellbeing and lower work outcomes. For example, Hirschi et al. [33] examined the within-individual change in the presence of a calling over one year. Their results suggest positive as well as negative effects of a calling on work–nonwork interface. In the same vein, more recently, Zhou et al. [34] have found that excessive levels of career calling were associated with detrimental workplace outcomes, such as work fatigue.

Third, the moderating effects of high work meaningfulness on the relationship between stress and mental health have recently been indicated. Firefighters with a high sense of calling experienced increased PTSD symptoms due to burnout compared to those with a low or an average calling [35]. Similarly, Wilson and Britt [65] found that the relationships between hindrance stressors and mental health symptoms were magnified when participants reported higher levels of calling.

Finally, it has been proposed that certain professions characterized by high levels of work meaningfulness (i.e., helping professions) may be especially vulnerable to its “dark side” when in an exploitive organizational environment [23]. Our starting point for exploring this proposition is the focus on the relationship between work meaningfulness and absenteeism among CM survivors employed in the helping professions.

Jung [66] coined the term “wounded healer” in describing the unconscious forces that drive CM survivors into the helping professions. While such a vocational choice may satisfy their inner need to heal other people as well as themselves, it puts a burden on survivors. Subsequent schema-focused models of occupational stress and work dysfunction [67] have suggested that individuals with early maladaptive schemes may be unconsciously drawn toward occupations with dynamics similar to the toxic environments and relationships they experienced in childhood [68]. Indeed, several studies have shown that the prevalence of CM among helping professionals is higher compared to the general population as well as in comparison to members of other professions [26,27,47].

High-functioning CM survivors often describe their careers in the helping professions as “callings” [29]. Many relate to their work as a strong source of significance and value in their lives by allowing them to serve as a role model, giving to others what they needed and never received, and contributing to a better world [28]. However, such a perception may come with a cost. Cowls and Galloway [69] claimed that some adult survivors of CM tend to be perfectionists and work excessively. Building on the literature regarding the “dark side” of a career calling, we suggest that alongside the benefits that such a perception may bring, CM survivors who work in the helping professions may also be uniquely sensitive to adverse impacts of high meaningfulness.

The literature on the nature of jobs in the helping professions shows that they are often characterized by high demands, including emotional pressures, working in shifts or on-call, a heavy workload, and work–home interference [36,37]. Moreover, working in a helping profession exposes employees to the traumatic experiences of clients, thus raising their risk of experiencing secondary traumatization [4,48]. For CM survivors, these traumatic events may echo their own pasts and may be experienced as a painful encounter [38]. Hence, due to their challenging pasts, working CM survivors may be more vulnerable, especially when employed in highly stressful environments or when experiencing traumatic job experiences. Some support for this notion comes from a recent study conducted among emergency medical service personnel, in which occupational trauma exposure was more strongly related to psychopathology among personnel who reported a higher exposure to CM [5].

The second aim of our study is to explore the moderating role of CM in the relationship between work meaningfulness and work absenteeism. We suggest that while work meaningfulness usually serves as a personal job resource for non-maltreated employees, for CM survivors who are helping professionals, work meaningfulness may be related to higher work absenteeism. Specifically, we hypothesize that CM will moderate the relationship between meaningfulness and absenteeism, so that non-maltreated professionals will report less absenteeism as they experience higher levels of work meaningfulness, while maltreated professionals will report more absenteeism as they experience higher levels of work meaningfulness (H2).

## 2. Materials and Methods

### 2.1. Sample and Procedure

High-functioning, working adults were recruited from a wide array of social and mental health services. To estimate sample size, G-power software (version 3.1.9.4) was used with the following statistical assumptions: a type 1 error of 5% and a minimum statistical power of 80% [70]. Based on previous studies, a moderate effect was expected between study variables [17,63], so the minimum sample size was estimated to be 209 respondents.

The original sample included 323 professionals recruited through email, social media, professional websites, and snowball sampling. Trained and credentialed research assistants administered questionnaires in-person to the participants who had signed informed consent forms. Participation was voluntary with no compensation received. The study was approved by the ethical review board of the second author’s academic institute (approval number 7019-30 L/sw).

Full demographic information was filled out by 320 participants. Participants’ ages ranged from 24 to 68 (*M* age = 40.7, *SD*  = 10.2) with 90.6% being female and 9.4% male. This ratio is consistent with the gender profile of helping professionals in Israel as well as that of helping professionals more widely [36,37]. In terms of marital status, the sample consisted of married/partnered (73.1%), single (19.2%), divorced/separated (6.2%), and widowed (1.2%) individuals. Furthermore, most participants identified Judaism as their religion (90.1%), while the rest identified Islam (6.5%), Christianity (2.2%), or other (4.6%). The mean number of years of education within the sample was found to be 16.94 (*SD* = 2.34), indicating a notably high level of educational attainment among participants. Regarding occupation, 62.2% of the sample were social workers, 14.2% were psychologists, psychotherapists, educational counselors, or speech therapists, and 19.2% were support staff (case managers, paraprofessionals, etc.). The average number of hours worked per week was 37.1 (*SD* = 10.5). The average number of years working in the current position was 6.64 (*SD* = 7.4), and the average number of years in the organization was 6.84 (*SD* = 7.5). Additionally, 22.6% of the participants reported having a managerial position, while 77.4% did not. The detailed demographic information for employees who reported CM and those who did not is provided in the Appendix A.

### 2.2. Measures

#### 2.2.1. Work Meaningfulness

The Hebrew-validated version of The Work and Meaning Inventory (WAMI) [56,71], comprised of 7 items (e.g., “I have found a meaningful career”), employing a unidimensional 7-point Likert-type scale (*strongly disagree* = 1, *strongly agree* = 7) was used. Cronbach’s alpha in our study was 0.79 and McDonald’s ω coefficient was 0.83.

#### 2.2.2. Work Absenteeism

One open-ended question was used: “In the last three months, how many days were you absent from your job (excluding scheduled holidays and vacation)?” Answers given by participants ranged between “0” and “12”. The mean was 2.25 days, the median was 2 days, and the mode was 0 days. A histogram depicting the frequency of absent days across the sample is provided in the Appendix A.

#### 2.2.3. Childhood Maltreatment History

The Adverse Childhood Experiences (ACEs) measure [72] is a 10-item measure assessing exposure to two domains of childhood adversities—“Childhood maltreatment” (5 items) and “Family/ household dysfunction” (5 items). Following Dube et al. [72], three categories of childhood abuse were used: emotional, physical, and sexual; and two categories of childhood neglect: emotional and physical, each represented by one item from the “childhood maltreatment” domain. For example, sample item is “While you were growing up, that is, in your first 18 years of life, did a parent or adult living in your home (1) swear at you, insult you, or put you down? (2) act in a way that made you afraid that you might be physically hurt?” (emotional abuse). As witnessing domestic violence and abuse is a distinct form of child maltreatment [73], another item from the “Family/ household dysfunction” dimension of the ACEs measure referring to witnessing violence between parents in childhood was also included.

Those responding “yes” to having undergone at least one of these forms of abuse or neglect were defined as having experienced CM [72]. After screening, it was determined that 111 participants (34.7%) had reported at least one type of abuse or neglect in childhood (CM survivors’ group, coded as 1), and 209 (65.3%) reported no CM (comparison group, coded as 0). The prevalence rates of childhood exposure to abuse and neglect by category are provided in the Appendix A.

## 3. Results

### 3.1. Descriptive Statistics and Correlations

Table 1 includes descriptive statistics and the correlation matrix of the study’s variables. CM history was positively correlated with absenteeism (*r* = 0.16, *p* = 0.00), providing preliminary support for H1. Other relationships were not significant.

### 3.2. Analyses of the Research Hypotheses

To test the first hypothesis, a t-test for independent samples was conducted comparing the number of absent days of employees who reported CM and that of those who did not. As can be seen in Figure 1, work absenteeism was significantly higher among maltreated employees (*M* = 2.82, *SD*  = 2.88) compared with their non-maltreated counterparts (*M* = 1.93, *SD*  = 2.45) (*t* = −2.91, *p* = 0.00, Cohen’s d = −0.34), thus supporting H1.

To fully test the two suggested hypotheses, Hayes’s [74] SPSS PROCESS macro (Model 1) with 95% bias-corrected bootstrapped confidence intervals (using 20,000 replications) was used. The dependent variable was work absenteeism (*Y*) and the independent variable was work meaningfulness (*X*). CM history was entered as possible moderator (*W*). Work meaningfulness (*X*) and CM history (*W*) were both centered. Gender, age, tenure, and managerial position (yes/no) were controlled for. Excluding participants with missing data, the final sample size used for model assessment was 255.

The model had 31% explained variance and was significant (*R*^2^ = 0.10, *F* (7,247) = 3.81, *p* = 0.00). The increase in the explained variance of the model between the model with the main effect only and the model with both the main effect and the interaction effect was 3% and was significant (Δ*R*^2^ = 0.03, *F* (1,247) = 7.77, *p* = 0.01).

Hypothesis 1 suggested that absenteeism levels among CM survivors will be higher compared to non-maltreated employees. The results (Table 2) showed that the effect of CM on work absenteeism was significant (*b* = 0.94, *p* = 0.01), supporting H1. Table 2 also indicates that work meaningfulness had no significant effect on work absenteeism (*b* = 0.13, *p* = 0.77).

Hypothesis 2 suggested that CM will moderate the relationship between meaningfulness and absenteeism, so that non-maltreated employees will report less absenteeism as they experience higher levels of work meaningfulness, while CM survivors will report more absenteeism as they experience higher levels of work meaningfulness. As shown in Table 2, a significant interaction effect between work meaningfulness and a history of CM was indicated (*b* = 2.50, *p* = 0.01). Moreover, being in a managerial position predicted less absenteeism. We found an average of 1.5 absent days for managers compared with an average of 2.5 days for non-managers.

An analysis of the simple slopes reveals that among employees with no history of CM, as work meaningfulness increased, work absenteeism decreased, yet this relationship was not significant (*b* = −77, *se* = 0.54, *t* = −1.47, *p* = 0.16, 95%*CI* (−1.84, 0.30). However, among employees with a history of CM, as work meaningfulness increased, work absenteeism increased as well, and this relationship was significant (*b* = 1.74, *se* = 0.71, *t* = 2.44, *p* = 0.02, 95%*CI* (0.33, 3.14) (see Figure 2).

## 4. Discussion

Our first finding suggests that a past of CM is positively related to work absenteeism in adulthood, providing additional support for the occupational and organizational costs of CM in adulthood [17,18,19]. One possible explanation for this association is the well-documented negative effects of childhood abuse and neglect on psychological and physical health in adulthood [41]. Specifically, experiencing CM has been predictive of mental health outcomes, such as anxiety and depression, and of various physical outcomes as well [42,43]. Indeed, De Venter and colleagues have found that current depressive disorders and current comorbid depression–anxiety have mediated the positive relationship between childhood trauma and adult work absenteeism [17]. Although not directly assessed in our study, maltreated employees may be more frequently absent from work than their non-maltreated counterparts due to coping with mood disorders or other health problems. Another possible explanation may be their increased vulnerability to work stress, such as higher acute stress responses and burnout [4,5,47,48]. Taking our results one step further, future studies should broaden the exploration of possible mediating mechanisms between a history of CM and work absenteeism, considering possible physical, mental, and occupational vulnerabilities.

Interestingly, managers in our study reported significantly lower absenteeism rates than non-managers. These results are in line with findings from previous studies [75,76]. A possible explanation may be the social expectations associated with their position [12]. Due to their influence over employee absence norms and associated absences, managers are expected to exhibit greater social responsibility regarding their attendance [77]. Additionally, research suggests that higher job titles are associated with greater job satisfaction [78]. An analysis of the Swedish Longitudinal Occupational Survey of Health (SLOSH) from 2010 found that managers had higher levels of job satisfaction and took fewer days off than non-managers [79].

Our second finding suggests that among employees with no history of CM, work meaningfulness was negatively associated with work absenteeism. Though this relationship was not statistically significant, it seems that for non-maltreated helping professionals, work meaningfulness may function as a personal resource, as suggested by the research focusing on meaningfulness and absenteeism among other types of professionals [56,59,63].

Our third finding questions the notion that work meaningfulness is unequivocally beneficial. In accordance with the second hypothesis, it was found that a history of CM moderated the relationship between work meaningfulness and absenteeism, so that among maltreated professionals, high work meaningfulness was positively related to work absenteeism. The positive and significant association discovered suggests that, for these employees, the perception of work as highly meaningful may be deleterious.

One possible mechanism that may explain these results is boundary inhibition [32]. Boundary inhibition refers to the mechanism in which high levels of work meaningfulness inhibit the adaptive regulation of boundaries around personal space. This challenge can also be framed as a difficulty in achieving psychological detachment, the ability to disengage oneself mentally when away from the workplace. It implies refraining from involvement in work-related tasks and not thinking about job-related issues when not at work [80].

Growing up in homes where boundaries were often blurred and mismanaged, CM survivors may struggle with boundary regulation in adulthood [9,81]. One may assume that when work is highly meaningful, CM survivors may participate more willingly in overwork and erratic work and be less inclined to disengage from their tasks. The result may be exhaustion, leading to more missed working days than among non-maltreated employees. As suggested by Jo et al. [35], a “Calling, though perceived as a positive variable, can be hazardous to exhausted people” (p. 117). Future studies should try to explore the possible mechanisms underlying the positive relationship between meaningfulness and absenteeism among CM survivors.

Our research contributes significantly to various theoretical streams within the existing literature. The positive association that was found between CM history and work absenteeism sheds light on one of the possible long-term occupational and organizational consequences of CM. Although there is substantial documentation regarding the detrimental impacts of a history of CM on survivors’ educational attainment, employment prospects, and income [1,2], as well as on their physical and mental health [41,42], knowledge of the underlying mechanisms that render them vulnerable or foster their resilience in the workplace remains limited [43,82]. Our study expands the current understanding of the work-related obstacles faced by these individuals and encourages further investigation into the interplay between personal and organizational resources and demands within this significant workforce segment.

The central role of CM history in explaining work absenteeism also follows recent advancements in the JD-Rs model, as advocated by Bakker and Demerouti [15] and Miraglia and Johns [12]. While heretofore the majority of research has focused on the influence of social, contextual, and organizational demands on work absenteeism [12,83], our study broadens the investigation into the impact of personal demands.

The third finding of your study is the potential drawback associated with work meaningfulness. This enhanced understanding of the “darker side” of highly meaningful work aligns with contemporary perspectives on the detrimental consequences of the inauthentic management of meaning by leaders and organizations, as discussed by Bailey et al. [84]. Our study underscores the necessity for a deeper exploration of the circumstances under which the cultivation of meaningful work can either benefit or potentially harm both individuals and organizations, as explored by scholars such as Lysova et al. [85]. Consequently, future research in the field of organizational behavior would benefit from embracing a more nuanced investigation of work meaningfulness outcomes.

### 4.1. Limitations and Future Research

This study has several limitations. First, it utilized a single source of data—a self-report survey. Future research employing a multi-source approach might clarify the identified patterns and offer additional, less-constrained information. Specifically, using human resource records with a fuller account for the reasons of absenteeism is highly recommended [86].

Second, we adopted a unified measure of work meaningfulness [56], while future studies may benefit from exploring the distinct roles of self-actualization vs. other-oriented work meaningfulness [57,58]. Moreover, we focused on the subjective conceptualization of work meaningfulness, which is based on the perception and preferences of the individual practicing it, not on the work itself [55,58]. Normative definitions, however, assert that meaningful work should hold significance for oneself and for others, and include the moral justification for why it is of value. Such definitions suggest criteria for discerning varying levels of meaningfulness in work, acknowledging the potential for misconceptions about one’s own work’s significance, such as morally commendable work lacking personal significance [87]. Broadening future explorations to include both subjective and normative definitions of work meaningfulness may help in better understanding the intricate associations between work meaningfulness and work outcomes [87]. In addition, future studies may benefit from exploring both work meaningfulness and a sense of calling, which, despite their similarities, represent distinct concepts [88].

Third, further research is needed to unravel the mechanisms that may explain the documented effects. These studies may consider processes, such as a diminished psychological distance [80], the exposure to traumatic experiences at work [38,69], and the development of secondary traumatization and burnout [4,48] as possible mediators.

Fourth, given the cross-sectional design of this study, causality cannot be inferred [89]. Longitudinal designs should be incorporated into future explorations of CM work-related outcomes. Finally, further investigation is needed as to whether the findings can be generalized to workers in other professions and to whether, since our survey was held prior to the COVID-19 pandemic, it is relevant to the post-COVID world.

### 4.2. Practical Implications

This study has several possible practical implications. First, leaders of teams including CM survivors should receive trauma-informed management training [48]. Leaders should be made aware of the potential double-edged nature of work meaningfulness and its implications for absenteeism. While promoting meaningful work is essential, leaders need to adopt a balanced approach. They should acknowledge that for some employees, an enhanced sense of work meaningfulness may not necessarily translate into better outcomes [22,84].

Second, at the organizational level, there is a need to raise the employee awareness of the potential impact of CM on work experiences and outcomes [90]. This can create a more empathetic and supportive work environment for CM survivors. Social and mental health organizations can also implement short-term absenteeism-reduction interventions, which have been found to be effective in lowering absenteeism rates among at-risk populations [91]. Moreover, organizations should consider reviewing and adapting their human relations policies and procedures to better accommodate the needs of CM survivors, ensuring they are not penalized for taking necessary leave. Another possible policy change would be to offer adjustable working arrangements (such as flexible hours and hybrid work), which may be especially beneficial for employees who have experienced CM [92]. These practical steps can serve as a starting point for addressing the complex connection between work meaningfulness, childhood maltreatment, and absenteeism among helping professionals.

## 5. Conclusions

While the positive aspects of engaging in meaningful work are widely recognized, there exists a relative dearth of knowledge regarding its potential adverse consequences, commonly referred to as the ‘dark side’ of meaningful work. The current research endeavors to scrutinize the assumption that an augmented sense of meaningfulness in the workplace invariably yields positive outcomes for individuals with a history of childhood maltreatment. Consistent with expectations, childhood maltreatment was found to be positively correlated with work absenteeism. Furthermore, our investigation unveiled a moderating effect of childhood maltreatment on the association between work meaningfulness and absenteeism. Among individuals with a history of maltreatment, heightened levels of perceived meaningfulness in their work correlated with increased occurrences of work absenteeism. This study underscores the nuanced nature of the relationship between work meaningfulness and its impact on employees, particularly those who are high-functioning survivors of childhood maltreatment, within organizational contexts. A comprehensive understanding of the challenges faced by this population is crucial for informing organizational policies and practices.

## Figures and Tables

**Figure 1 ijerph-21-00451-f001:**
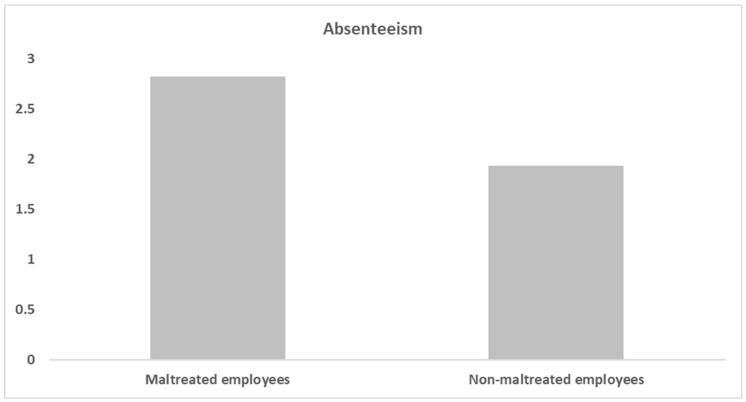
Absenteeism among maltreated and non-maltreated employees.

**Figure 2 ijerph-21-00451-f002:**
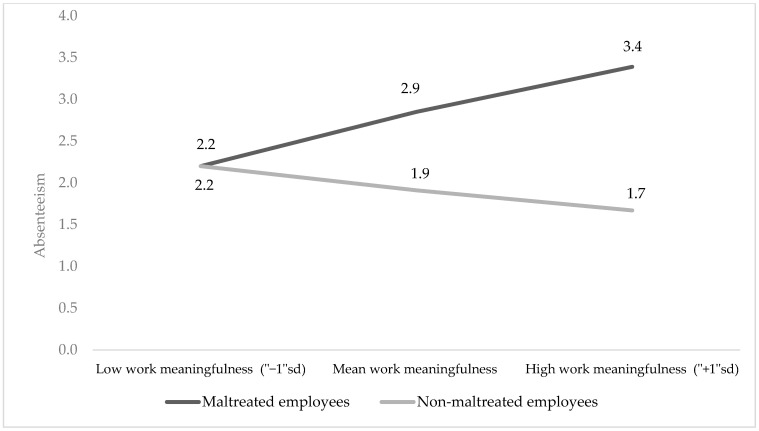
Interaction effect of CM and work meaningfulness on work absenteeism.

**Table 1 ijerph-21-00451-t001:** Means, standard deviations, and correlations of the study’s variables.

	Variable	*M*	*SD*	1	2
1	CM	0.36	1.48	-	
2	Work meaningfulness	4.68	0.4	0.04	-
3	Work absenteeism	2.25	2.65	0.16 **	−0.04

Note. *M* and *SD* are used to represent mean and standard deviation, respectively. ** *p* < 0.01.

**Table 2 ijerph-21-00451-t002:** Results of hypothesis testing.

	Variable	*B*	*SD*	*T*	*LLCI*	*ULCI*
1	CM	0.94 *	0.34	2.81	0.28	1.6
2	Work meaningfulness	0.13	0.43	0.29	−0.72	0.97
3	CM X Meaningfulness	2.50 *	0.9	2.79	0.73	7.27
4	Managerial position	0.89 *	0.39	2.25	0.11	1.66
5	Tenure	0.06	0.03	2	0	0.11
6	Age	0.04	0.02	1.95	0	0.08
7	Gender	0.15	0.56	0.27	−0.96	1.26
8	Constant	−85.43	43.06	−1.94	−168.26	1.36

Note. *N* = 255. The dependent variable is work absenteeism. Regarding CM, those with one or more “yes” were coded as 1 (“CM group”); the rest were coded as 0. Being in a managerial position was coded as 1 = no, 2 = yes; Gender was coded as 1 = female, 2 = male. * *p* < 0.05.

## Data Availability

The data can be obtained by mail from the corresponding author icekson@bgu.ac.il.

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
