# Peer review of "Childhood Maltreatment and Adult Work Absenteeism: Work Meaningfulness as a Double-Edged Sword"

_ijerph, 2024, doi:10.3390/ijerph21040451_

Round 1

Reviewer 1 Report

Comments and Suggestions for Authors

This very interesting short and simple paper offers a fresh, innovative perspective on work meaningfulness, proposing its complex (and potentially negative) effects. The integrative theoretical understanding that the paper offers and its findings can hold important theoretical and practical implications for work meaningfulness, absenteeism, and long-term effects of child maltreatment. Furthermore, the understandings that the manuscript provide can perhaps be relevant for related phenomenon such as burnout and turnover, and raise our awareness to other vulnerable employee groups (in various professions) – thus they may trigger additional research to explore related/similar phenomena.

The paper is concise, clear, and well-written. The abstract clearly conveys the study’s rationale and theoretical framework, and includes all the needed information about the methods, findings, and implications. The literature review is concise and provides a profound theoretical and empirical basis for the study’s framework and hypotheses. The method is suitable for examining the study’s hypotheses, and the method section clearly describes the participants, tools and procedure. The data analysis using PROCESS is appropriate for examining the hypotheses, and it is clearly described in a reader-friendly way. I have only praise and no comments about these sections.

The discussion provides a clear summary of the findings and their theoretical implications. Importantly, the authors suggest the mechanism of boundary inhibition and the difficulty in achieving psychological detachment as relevant mechanisms that may underlie the increased absenteeism of employees with CM. This is a very thoughtful direction, that can trigger important additional research. The authors also propose practical implications and suggestions for managers and organizations, based on the study’s findings. Most of these include revealing the CM history and discussing it with the organization or the manager. I think this strategy should be considered with cautious, as it may also have negative implications for the employee (in the short or long term).

Overall, this innovative, well-written manuscript can clearly contribute to the literature, and in my opinion, should be accepted with minor revisions.

Author Response

We highly appreciate the reviewer’s feedback. We agree that revealing a painful history to the manager or the organization may yield negative results. We have therefore deleted this suggestion and focused more on managerial and organizational implications (see page 13, lines 498-503, page 14, lines 504-514).

Reviewer 2 Report

Comments and Suggestions for Authors

These are my comments:

1. The abstract particularly in the beginning needs to be improved. It is vague and misdirected.

2. What is the study motivation, objectives and aims? it is not clearly stated.

3. 1.3 should be rephrased. Do not mention moderator. This should be explained in hypotheses development.

4. 1.4, this is unacceptable. It should be embedded within the hypotheses development. It should be developed within 1.1-1.3

5. Result. Where is the demographic profiles? The descriptive statistics?

6. There are many instances in the texts seems to have different color font. Please check

7. There is no reference to past work in practical implications.

8. What about theoretical implications?

9. The structure of the paper needs to be aligned.

10. The paper would benefit from proofreading.

Comments on the Quality of English Language

It can be accepted after major revision.

Author Response

Reply to comments made by reviewer 2(R2)

These are my comments:

  1. The abstract particularly in the beginning needs to be improved. It is vague and misdirected.

Response:

We appreciate the feedback and have carefully revised the beginning portion of the abstract to address any vagueness and ensure clarity in our presentation. Our revisions aim at enhancing its overall quality and coherence. Please see page 1, lines 14-21.

  1. What is the study motivation, objectives and aims? it is not clearly stated.

Response:

In response to this valuable feedback, we have revised the manuscript to explicitly outline the study's objectives and aims in the Introduction section. We believe the changes we have made will provide a clearer understanding of the study's purpose and direction. Please see page 2, lines 41-47; page 3, lines 100-102;  page 4, lines 163-164; page 5, lines 165-166, page 7, lines 259-263.

  1. 1.3 should be rephrased. Do not mention moderator. This should be explained in hypotheses development.

Response:

Following this important comment, and as part of our effort to improve the flow of the theoretical background, there are now two parts to the first section,  and their subtitles have been renamed; 1.1 is now “Child maltreatment and absenteeism” and 1.2 is “Work meaningfulness as a double-edged sword: The role of child maltreatment history.” We believe these amendments provide a clearer understanding of the study's  building  of hypotheses.

  1. 1.4, this is unacceptable. It should be embedded within the hypothesis’s development. It should be developed within 1.1-1.3

Response:

We agree with this comment and have embedded the development of the study’s hypotheses within 1.1-1.2. Section 1.4 has been omitted from the current version of the manuscript (see page 7).

  1. Result. Where is the demographic profiles? The descriptive statistics?

Response:

Demographic information regarding the sample appears on page 7 (lines 281-290). In addition, we have now added an appendix with fuller detailed demographic information regarding maltreated and the non-maltreated samples (see supplementary materials file).

  1. There are many instances in the texts seems to have different color font. Please check

Response:

The text of the manuscript we are submitting has one color font.

  1. There is no reference to past work in practical implications.

Response:

In response to this important comment, relevant references were added to the practical implications section (see page 13, lines 499, 503, 505, 508, 512).

  1. What about theoretical implications?

Response:

The three main theoretical contributions of the current study are now better elaborated towards the end of the discussion,  on page 12, lines 445-461 and on page 13, lines 462-468.

  1. The structure of the paper needs to be aligned.

Response:

The structure of the paper has been aligned.

  1. The paper would benefit from proofreading.

Response:

Our revised manuscript has been carefully proofread.

In sum, we would like to thank the editor and the reviewers, who have clearly put great effort into carefully reading the manuscript and have offered us insightful suggestions which have been extremely helpful in improving it. We look forward to your feedback on our revised version.

Reviewer 3 Report

Comments and Suggestions for Authors

This study examined the relationship between work meaningfulness and work absenteeism among helping profession workers and further explored the moderation role of child maltreatment history. The results showed that the relationship between work meaningfulness and work absenteeism is not significant, and only when child maltreatment history is considered did the study reveal a significant interaction effect. It is shown that work meaningfulness could positively predict absenteeism for workers with a child maltreatment history, and for those without such history, work meaningfulness has no significant relation with absenteeism. The research topic is important, but several issues should be addressed. 

1.    The research question is not properly defined. From the research design, it seems that the authors are more interested in the impact of child maltreatment history on absenteeism, but the introduction gives one an impression that the main research question is about work meaningfulness (and its “dark side”). 

2.    The Introduction defined “work meaningfulness” on Page 2, but the sudden transition to its negative aspect is abrupt and lacks sufficient justification. 

3.    In the moderation analysis, were the two variables that make up the interaction term centered?

4.    The core variable, work absenteeism, is measured by number of absent days. Because it is the focus of the study, more descriptive statistics on this variable are needed, such as median, mode, or histogram. 

5.    Would it be more direct to test whether CM could impact absenteeism by simply conducting a t-test comparing absent days of participants with vs. without CM? I think the t-test will be significant as the correlation in Table 1 is significant, but it would be clearer to list the absent days of participants with or without CM in a table or a bar plot, together with a t-test result. 

6.    In the regression table (Table 2), it is notable that aside from the focal variables of the study, there is another significant predictor, “Managerial Position”. This result suggests that being in a managerial position alone could account for nearly one more day (0.89) absent than regular workers. Why? 

Author Response

Reply to comments made by reviewer 3 (R3)

  1. The research question is not properly defined. From the research design, it seems that the authors are more interested in the impact of child maltreatment history on absenteeism, but the introduction gives one an impression that the main research question is about work meaningfulness (and its “dark side”). 

Response:

We thank the reviewer for bringing this issue to our attention. We recognize the importance of clearly defining the primary focus of our study. Our first research question indeed revolves around investigating the impact of CM history on work absenteeism, as indicated by the research design. Our second research question investigates the moderating role of CM history on the relationship between work meaningfulness and work absenteeism. However, we acknowledge that the introduction may have given the impression that the negative implications of work meaningfulness are the primary emphasis. To rectify this, we have revised the opening part of the abstract (see page 1, lines 14-22),  and the introduction (see page 1 lines 33-40; page 2, lines 41-47 and lines 79-81; page 3, lines 82-102). In addition,  we have reorganized the literature review and rewritten parts of it (see page 3, lines 105-123; page 4, lines 124-164; page 5, lines 165-199). By making these adjustments, we aim to ensure alignment between the stated research questions and the overarching objectives of the study.

  1. The Introduction defined “work meaningfulness” on Page 2, but the sudden transition to its negative aspect is abrupt and lacks sufficient justification. 

Response:

An effort was made to improve the clarity and coherence of the introduction of “work meaningfulness” as a significant variable in the study. Please see page 2, lines 73-81, page 3, lines 82-102.

  1. In the moderation analysis, were the two variables that make up the interaction term centered?

Response:

In response to this comment, yes, the variables were centered when running the model in PROCESS. We have added a clarification on page 9, line 359; page 10, line 360.

  1. The core variable, work absenteeism, is measured by number of absent days. Because it is the focus of the study, more descriptive statistics on this variable are needed, such as median, mode, or histogram. 

Response:

We agree with this important comment.  We have added median and mode calculations to the manuscript (page 8, lines 308-310). In addition, a histogram representing the frequency of absent days in the sample was added to the submission (see supplementary materials file, page 3).

  1. Would it be more direct to test whether CM could impact absenteeism by simply conducting a t-test comparing absent days of participants with vs. without CM? I think the t-test will be significant as the correlation in Table 1 is significant, but it would be clearer to list the absent days of participants with or without CM in a table or a bar plot, together with a t-test result. 

Response:

Following the reviewer’s suggestion, a t-test comparing absent days of participants with vs. without CM was conducted. As suggested, the results are reported on page 9 (lines 348-352) and are illustrated in figure 1 on page 9.

  1. In the regression table (Table 2), it is notable that aside from the focal variables of the study, there is another significant predictor, “Managerial Position”. This result suggests that being in a managerial position alone could account for nearly one more day (0.89) absent than regular workers. Why? 

Response:

We thank the reviewer for bringing this important question to our attention. In our revision  of the manuscript,  we clarify this finding in the results section on page 10, lines 376-378. In addition, we briefly discuss this finding in the light of previous literature on page 11, lines 412-419.

In sum, we would like to thank the editor and the reviewers, who have clearly put great effort into carefully reading the manuscript and have offered us insightful suggestions which have been extremely helpful in improving it. We look forward to your feedback on our revised version.

Round 2

Reviewer 2 Report

Comments and Suggestions for Authors

Thank you the authors for revising the manuscript. I believe now the manuscript is suitable for publication.